Computational analysis of functional SNPs in Alzheimer’s disease-associated endocytosis genes

Tey Han Jieh
Ng Chong Han chng@mmu.edu.my
Faculty of Information Science and Technology, Multimedia University , Ayer Keroh , Melaka , Malaysia
Qin Zhaohui
Electronic publication date: 2019 Sep 30
Publication date: 2019
Volume: 7
Electronic Location ID: e7667
Received 2019 Mar 21; Accepted 2019 Aug 13
Copyright: ©2019 Tey and Ng
Copyright year: 2019
Copyright holder: Tey and Ng
License: This is an open access article distributed under the terms of the Creative Commons Attribution License, which permits unrestricted use, distribution, reproduction and adaptation in any medium and for any purpose provided that it is properly attributed. For attribution, the original author(s), title, publication source (PeerJ) and either DOI or URL of the article must be cited.
License URL: https://creativecommons.org/licenses/by/4.0/

Keywords: SNP prediction, Alzheimer’s disease, Computational analysis, Endocytosis genes, PICALM, SH3KBP1, SYNJ1

Funding: Fundamental Research Grant Scheme MMUE.140076 Malaysia Ministry of Higher Education (MOHE) This research is supported by the Fundamental Research Grant Scheme (grant number: MMUE.140076) from Malaysia Ministry of Higher Education (MOHE). The funders had no role in study design, data collection and analysis, decision to publish, or preparation of the manuscript.

==============================
Background

From genome wide association studies on Alzheimer’s disease (AD), it has been shown that many single nucleotide polymorphisms (SNPs) of genes of different pathways affect the disease risk. One of the pathways is endocytosis, and variants in these genes may affect their functions in amyloid precursor protein (APP) trafficking, amyloid-beta (Aβ) production as well as its clearance in the brain. This study uses computational methods to predict the effect of novel SNPs, including untranslated region (UTR) variants, splice site variants, synonymous SNPs (sSNPs) and non-synonymous SNPs (nsSNPs) in three endocytosis genes associated with AD, namely PICALM, SYNJ1 and SH3KBP1.

Materials and Methods

All the variants’ information was retrieved from the Ensembl genome database, and then different variation prediction analyses were performed. UTRScan was used to predict UTR variants while MaxEntScan was used to predict splice site variants. Meta-analysis by PredictSNP2 was used to predict sSNPs. Parallel prediction analyses by five different software packages including SIFT, PolyPhen-2, Mutation Assessor, I-Mutant2.0 and SNPs&GO were used to predict the effects of nsSNPs. The level of evolutionary conservation of deleterious nsSNPs was further analyzed using ConSurf server. Mutant protein structures of deleterious nsSNPs were modelled and refined using SPARKS-X and ModRefiner for structural comparison.

Results

A total of 56 deleterious variants were identified in this study, including 12 UTR variants, 18 splice site variants, eight sSNPs and 18 nsSNPs. Among these 56 deleterious variants, seven variants were also identified in the Alzheimer’s Disease Sequencing Project (ADSP), Alzheimer’s Disease Neuroimaging Initiative (ADNI) and Mount Sinai Brain Bank (MSBB) studies.

Discussion

The 56 deleterious variants were predicted to affect the regulation of gene expression, or have functional impacts on these three endocytosis genes and their gene products. The deleterious variants in these genes are expected to affect their cellular function in endocytosis and may be implicated in the pathogenesis of AD as well. The biological consequences of these deleterious variants and their potential impacts on the disease risks could be further validated experimentally and may be useful for gene-disease association study.

Introduction

Alzheimer’s disease (AD) is the most common type of dementia. According to the amyloid hypothesis, amyloid-beta (Aβ) is the primary factor to initiate a pathogenic cascade of AD in the brain (see review in Du, Wang & Geng (2018)). Aβ is generated from proteolysis of its precursor, amyloid precursor protein (APP). The oligomeric assemblies of Aβ that form amyloid plaques in extracellular space of neuron cell cause synaptic loss between neurons. Endocytosis is one of the biological pathways affecting APP trafficking and many endocytosis genes, including BIN1, CD2AP, PICALM, EPHA1 and SORL1 were identified as AD-associated in genome wide association study (GWAS) and other genetic studies (see review in Giri, Zhang & Lü (2016)).

From a study by Treusch et al. (2011), 40 types of Aβ modifiers were identified through a yeast genome-wide overexpression study, which demonstrated a genetic link between Aβ and endocytosis pathway in the pathogenesis of AD. There are two types of Aβ modifiers, namely Aβ suppressors, which decrease the toxicity of soluble Aβ oligomers and Aβ enhancers, which increase the toxicity. Among the identified Aβ modifiers, three of the Aβ suppressors are involved in endocytosis, namely YAP1802, INP52 and SLA1, and they have human orthologues, PICALM, SYNJ1 and SH3KBP1, respectively. Orthologous genes typically have similar biological function. Therefore, these three human orthologue genes may also play a role in modifying the toxicity of Aβ oligomers in human. Besides that, these human orthologues also play a role in clathrin-mediated endocytosis (CME), which is the major route for synaptic vesicle recycling (see review in Milosevic (2018)). In addition, CME and its regulators have also been linked to the trafficking of Aβ (Domínguez-Prieto et al., 2018). Therefore, it is of great interest to study the potential biological roles of PICALM, SYNJ1 and SH3KBP1 genes in the pathogenesis of AD.

PICALM (Phosphatidylinositol binding clathrin assembly protein) is mostly expressed in the brain capillary endothelium and neuron cells, and it is involved in cellular trafficking and the formation of clathrin-coated pit during CME (Tebar, Bohlander & Sorkin, 1999). PICALM gene is one of the validated AD risk genes first identified in a GWAS (Harold et al., 2009). The increased level of PICALM protein promotes the internalization of APP while its reduction causes the translocation of APP on the cell surface (Xiao et al., 2012). Besides that, PICALM helps in Aβ clearance by increasing Aβ translocation across the blood–brain barrier (Zhao et al., 2016). Several PICALM variants, including rs3851179 and rs592297, were identified to be associated with AD (Harold et al., 2009).

SYNJ1 (Synaptojanin 1) is the major synaptic polyphosphoinositide phosphatase, which regulates the uncoating and remobilization of synaptic vesicles during CME at the axon terminus (Cremona et al., 1999). SYNJ1 interacts with one of the AD risk genes, BIN1 (Bridging integrator 1) (Ramjaun et al., 1997). ApoE4-induced upregulation of SYNJ1 leads to the phospholipid homeostatic dysregulation in the brain, which is associated with the cognitive deficits in AD (Zhu et al., 2015). Conversely, downregulation of SYNJ1 not only restores the phospholipid homeostasis, but also reduces the level of phosphorylated tau protein in vitro (Cao et al., 2017). On the other hand, variant in SYNJ1 gene was found to be associated with the heritable form of Parkinson’s disease (PD) (Krebs et al., 2013).

SH3KBP1 (SH3-Domain Kinase Binding Protein 1), also known as CIN85, is an adaptor protein with multiple cellular functions through binding to different protein partners. SH3KBP1 is a homologous gene to one of the validated AD risk genes, CD2AP (see review in Rosenthal & Kamboh (2014)). These two homologous genes were reported to have high sequence and structural similarities, and they were identified as interacting partners (Mutso et al., 2018). A transgenic Drosophila study also showed the reduction of cindr gene expression, a fly orthologue of SH3KBP1, increased tau neurotoxicity (Shulman et al., 2014).

Single nucleotide polymorphisms (SNPs) play important roles in many complex human diseases, including auto-immune diseases, diabetes and schizophrenia (see review in Visscher et al. (2017)). SNPs that fall on either coding or non-coding regions of a genome can exert their biological effects and could cause diseases. There are two types of SNPs in the coding regions, namely synonymous SNPs (sSNPs) and non-synonymous SNPs (nsSNPs). For sSNPs, they do not change the amino acid sequence of a translated protein. However, they can exert their biological consequences through changes in alternative splicing, mRNA stability or translation rate (see review in Sauna & Kimchi-Sarfaty (2011)). While for nsSNPs, these variants cause amino acids substitution and may have direct structural and functional impacts to the proteins. Non-coding regions SNPs may affect transcription factor binding, gene splicing, RNA degradation and other biological processes.

Previous studies have demonstrated the important roles of genetic variants in the development of AD. However, there are limited studies about the functional variants in these AD-associated endocytosis genes, especially SYNJ1 and SH3KBP1 genes. In this study, the effect of UTR variants, splice site variants, sSNPs and nsSNPs in PICALM, SYNJ1 and SH3KBP1 genes, was predicted and analyzed using computational methods. The variants with high probability of affecting the protein structure and function are identified in the present study.

Material and Methods

Dataset retrieval

The information of the genes of interest was retrieved from the Ensembl genome database (release 87) (http://www.ensembl.org/index.html) in March 2017 for computational analysis. Ensembl comprises the variant data from multiple sources, such as dbSNP, COSMIC and DVGa. The positions of the variants were mapped on the human reference genome GRCh38, which is also equivalent to the UCSC (University of California, Santa Cruz) hg38.

Analysis of UTRs variants

Both 3′and 5′ UTR sequences of the genes of interest were collected from the curated database, UTRdb (http://utrdb.ba.itb.cnr.it/). These sequences with different variants were submitted to the UTRScan, which is a pattern matcher to search for UTR motifs collected in UTRSite (Grillo et al., 2010). UTRSite is a collection of functional sequence patterns located in 3′ and 5′ UTR sequences, such as upstream open reading frame (uORF), internal ribosome entry site (IRES), and polyadenylation signal (PAS). If different functional patterns are found between wild type and mutant UTR sequences, the variant is predicted to be functionally significant.

Analysis of splice site variants

The impact of splice site variants was predicted by using MaxEntScan integrated in Ensembl Variant Effect Predictor (VEP) (http://www.ensembl.org/Tools/VEP), which implements the Maximum Entropy Distribution approach. The input for the VEP requires only variant identifier (variant ID). For each variant, a consensus score with value in the range of −20 to +20 is calculated by MaxEntScan. Then, the score difference (in percentage) between wild type and mutant is calculated manually. The consensus score threshold of 3 and score difference threshold of 30% were applied to predict the splice site variants from both positive and negative controls (Desmet et al., 2009). Table S1 shows the possible consequences of splice site variants predicted by MaxEntScan.

Analysis of sSNPs

PredictSNP2 (http://loschmidt.chemi.muni.cz/predictsnp2/) is a web server that predicts the effect of nucleotide substitution in any region of the genome. The server integrates five prediction tools—CADD, DANN, FATHMM, FunSeq2 and GWAVA (Bendl et al., 2016). The consensus score generated by PredictSNP2 ranges from −1 to +1, in which the variant is considered neutral if the consensus score falls between −1 and 0, and is deleterious if the consensus score falls between 0 and +1.

Analysis of nsSNPs

Five prediction software packages that implement different prediction methodologies, namely SIFT (Ng & Henikoff, 2001), PolyPhen-2 (Adzhubei, Jordan & Sunyaev, 2013), Mutation Assessor (Reva, Antipin & Sander, 2011), I-Mutant2.0 (Capriotti, Fariselli & Casadio, 2005) and SNPs&GO (Capriotti et al., 2013), were used to predict the deleteriousness of all the nsSNPs. The nsSNPs that passed all the default cutoff points were considered to be deleterious nsSNPs. Then, the degree of evolutionary conservation of the deleterious nsSNPs was estimated using the ConSurf server (Ashkenazy et al., 2016), and the mutant protein structures were modelled and refined by SPARKS-X (Yang et al., 2011) and ModRefiner (Xu & Zhang, 2011).

SIFT (Sorting Intolerant From Tolerant, version 5.2.2) from http://sift.jcvi.org/ is a sequence conservation-based package that predicts the functional impact of amino acid substitution based on sequence homology and physiochemical diversity (Ng & Henikoff, 2001). The cutoff point was set at 0.05, so variants with a SIFT score lower than 0.05 were considered deleterious.

PolyPhen-2 (Polymorphism phenotype 2, version 2.2.2) from http://genetics.bwh.harvard.edu/pph2/, is a sequence- and structure-based prediction package. It predicts the impact of nsSNPs based on sequential and structural features collected from multiple databases and tools, such as UniProtKB, BLAST, Pfam and DSSP (Adzhubei, Jordan & Sunyaev, 2013). The cutoff point was set at 0.9, so variants with PolyPhen-2 score (known as PSIC score) higher than 0.9 were categorized as probably damaging.

Mutation Assessor (release 3) from http://mutationassessor.org/r3/ performs the sequence conservation-based prediction by assessing the entropy differences of an alignment position. It uses the sequence homology information of both protein families and subfamilies to assess the level of conservation of each residue (Reva, Antipin & Sander, 2011). The functional impact of the variants was categorized into “neutral”, “low”, “medium” or “high” impact categories. In this study, the cutoff level for the deleterious variant was set at “medium” or “high” impact.

I-Mutant2.0 (http://folding.biofold.org/i-mutant/i-mutant2.0.html) is a structure analysis-based prediction software that analyzes the protein stability change for a single point variant (Capriotti, Fariselli & Casadio, 2005). I-Mutant2.0 works by calculating the change in Gibbs free energy between the wild type and mutant protein structures. Variants with energy change value (DDG) less than −1 or greater than 1 indicate that these variants are destabilizing or stabilizing the protein, respectively.

SNPs&GO (Single Nucleotide Polymorphism Database & Gene Ontology), at http://snps.biofold.org/snps-and-go/snps-and-go.html, combines both sequence and structure information to perform prediction. Information from Gene Ontology terms is also taken into consideration in predicting the functional impact of the variants (Capriotti et al., 2013). The cutoff point was set at 0.5, so variants with probability greater than 0.5 were predicted as disease-related.

ConSurf (http://consurf.tau.ac.il/2016/) is a widely used evolutionary conservation analysis tools based on amino acid positions among the homologous protein sequences (Ashkenazy et al., 2016). In ConSurf, the conservation score of each protein residue indicates the level of conservation, with score “1” indicating the residue is highly variable and score “9” indicating the residue is highly conserved. Residues are also predicted to be buried (b), exposed (e), highly conserved and exposed (functional, f), or highly conserved and buried (structural, s).

SPARKS-X fold recognition server (http://sparks-lab.org/yueyang/server/SPARKS-X/index.php) is a template-based protein structure modelling tool (Yang et al., 2011). The modelled protein structures were refined by using ModRefiner, which is a structure refinement server that improves the structure quality, in terms of side-chain positioning and backbone topology (Xu & Zhang, 2011). The structural impacts of the nsSNPs can be predicted by studying the total free energy changes between the wild type and mutant protein structures, as well as the root mean square deviation (RMSD) of the mutant structures from the native protein structure.

Results

Dataset retrieval from Ensembl

The SNP datasets of PICALM, SYNJ1 and SH3KBP1 genes were retrieved from the Ensembl genome database. It contained a total of 1741 SNPs, including 395 UTRs (22%) variants, 188 (11%) splice site variants, 399 (23%) sSNPs and 759 (43%) nsSNPs. Out of a total of 759 nsSNPs in the genes of interest, 106 were classified as “deleterious” in SIFT and “probably damaging” in PolyPhen-2, according to the results in the Ensembl genome database. Figure S1 shows the total number of variants used in this study.

Prediction analysis of UTRs variants

The information of the retrieved UTR sequences and the total number of variants on the corresponding region are shown in Table S2. There were no 5′ UTR variants in SYNJ1 gene in the Ensembl genome database.

In the wild type UTR sequences of PICALM gene, a total of 11 motifs were identified from the UTRScan. Out of a total of 35 UTR variants in PICALM gene, only one 3′  UTR variant, rs796143140, caused an additional motif, namely K-box at position 212–219 of 3′ UTR sequence. K-box is one of the 3′ UTR motifs that mediates the negative post-transcriptional regulation of RNA (Ganguli et al., 2013). K-box motif was not present in the wild type UTR sequence of PICALM gene and the addition of the motif may affect transcript turnover and translational efficiency.

In the wild type 3′ UTR sequence of SYNJ1 gene, a total of 20 motifs were identified from UTRScan. Three 3′ UTR variants, including rs777138831, rs147590143 and rs111516740, caused an addition of upstream open reading frame (uORF) while another four variants, including rs767649429, rs143739621, rs761852713 and rs79652470, resulted in a deletion of uORF in 3′ UTR. Variant rs780084162 caused an additional motif named UNR-binding site (UNR-bs). UNR is one of the RNA-binding proteins that acts as a regulator in the initiation of internal ribosomal entry site (IRES)-mediated translation and cap-dependent translation (Saltel et al., 2017). Therefore, variant rs780084162 may affect the expression of SYNJ1 gene and the actual biological effect of the additional UNR-bs needs to be further experimentally validated.

For SH3KBP1 gene, there were a total of 21 motifs identified in the wild type 3′ and 5′ UTR sequences. Variant rs192424738 caused an addition of uORF; variant rs755895016 caused a deletion of uORF in the 3′ UTR sequence. Variant rs778448080 resulted in a deletion of GY-box at position 2064–2070 of the 3′ UTR sequence. Like K-box and Brd-box, GY-box is one of the 3′ UTR motifs that mediates the negative post-translational regulation of RNA (Lai, Tam & Rubin, 2005). Therefore, deletion of the GY-box caused by variant rs778448080 may affect the translation efficiency of SH3KBP1 gene.

Table 1 lists the variants that change the number of motif matched in UTRSite, as compared with their wild type UTR sequences (see Table S3 for complete prediction analysis results). Both 3′and 5′UTRs are enriched with cis-acting regulatory elements, and both UTRs are important for the transcriptional and translational regulation. In this study, a total of 12 UTR variants were predicted to cause an addition or a deletion of regulatory elements in the UTR sequences. Further analysis on the impacts of these regulatory elements, including the effect of the miRNA binding to the UTR sequence is outside the scope of this study. However, it should be characterized in the future.

Table 1 Variants with altered functional pattern(s) of the UTRs of PICALM, SYNJ1 and SH3KBP1 genes.

Variants ID	Ref/Alt	Minor allele frequency	Number of motif	Remarks	
			3′	5′		
PICALM						
Wild type			10	1		
rs796143140	T/C	NA	11	1	Addition of K-box at position 212-219	
SYNJ1						
Wild type			20	–		
rs777138831	G/C	NA	21	–	Addition of uORF at position 1915-1995	
rs147590143	G/T	NA	21	–	Addition of uORF at position 1915-1995	
rs780084162	G/T	NA	21	–	Addition of UNR-bs at position 1037-1047	
rs767649429	-/AA	NA	19	–	Deletion of uORF at position 942-1037	
rs143739621	G/A	0.00002/2	19	–	Deletion of uORF at position 942-1037	
rs111516740	G/A	0.0004/54	21	–	Addition of uORF at position 887-979	
rs761852713	-/G	0.0001/7	17	–	Deletion of uORFs at position 372-437, 464-535 and 644-751	
rs79652470	C/T	0.0001/13	17	–	Deletion of uORFs at position 182-289, 372-437, 464-535, 644-751, addition of uORF at position 205-810	
SH3KBP1						
Wild type			19	2		
rs778448080	G/T	NA	18	2	Deletion of GY-box at position 2064-2070	
rs192424738	C/T	NA	20	2	Addition of uORF at position 1380-1469	
rs755895016	A/-	NA	18	2	Deletion of uORF at position 1299-1370	

Prediction analysis of splice site variants

The prediction analysis of splice site variants was done using MaxEntScan, which is integrated in VEP. Submission to MaxEntScan requires only the variant IDs. The consensus score and score difference between wild type and mutant sequences were obtained after the submission. The consensus score for each variant was calculated based on different protein-coding transcripts and the same variant may have different consensus scores on different transcripts. This allowed users to study the impact of the splicing variant on different transcripts. Table 2 shows the variants with score difference exceeding the defined threshold. Columns “MaxEntScan Ref” and “MaxEntScan Alt” indicate the wild type and mutant consensus scores, respectively. “Score difference” shows the differences between the wild type and mutant consensus scores (as percentages) to predict the variant effects in causing exon extension, intron retention or splice site creation. Table S4 comprises the complete prediction analysis results for all the splice site variants.

Table 2 List of variants predicted to break or create splice site of PICALM, SYNJ1 and SH3KBP1 genes.

Variants ID	Ref/Alt	Minor allele frequency	Exon	Intron	MaxEntScanRef	MaxEntScanAlt	Score difference (%)	
PICALM								
rs371001564	G/A	0.00002/2	15/20		9.065917	0.623344	−93.12	
rs781144984	T/C	0.000008/1	16/20		7.971843	2.847240	−64.28	
rs768239913	A/G	0.00007/8	18/20		8.542122	2.867856	−66.43	
SYNJ1								
rs779813770	C/A	0.00009/1		1/32	8.703310	0.199354	−97.71	
rs376825246	T/C	0.000008/1		11/32	7.844178	3.885060	−50.47	
rs756697570	C/T	0.00005/6	12/33		7.638176	5.268185	−31.03	
rs748110438	G/T	0.00003/3		13/32	8.681559	0.239421	−97.24	
rs367718431	T/G	0.000009/1		13/32	8.681559	5.646822	−34.96	
rs367718431	T/C	0.000009/1		13/32	8.681559	4.215421	−51.44	
rs758644105	G/T	0.000008/1		16/32	8.941208	0.759180	−91.51	
rs746368700	A/T	0.000009/1	17/33		9.256318	4.683645	−49.40	
rs538934316	C/T	0.00003/4		19/32	5.263433	−3.346463	−163.58	
rs755765674	A/C	0.00002/2	23/33		6.505994	3.822270	−41.25	
rs533064963	T/A	0.00002/3		25/32	2.515213	3.906733	+55.32	
rs61752550	G/A	0.0001/11	29/33		9.393973	3.084793	−67.16	
rs373831795	T/A	NA		31/32	10.077445	1.573490	−84.39	
SH3KBP1								
rs777396528	T/C	0.00001/1		8/17	8.138004	4.452207	−45.29	
rs61761898	G/C	0.0035/303	15/18		9.095315	5.850377	−35.68	

A total of 18 variants were predicted to either disrupt or create splice sites, including 3, 13 and 2 variants from PICALM, SYNJ1 and SH3KBP1 genes, respectively. Among these variants, 17 were predicted to disrupt the splice site, in which splicing was omitted or disrupted, resulting in exon extension or intron retention. Variant rs533064963 in SYNJ1 gene was the only variant predicted to create a new splice site in the intron.

Genetic variations are known to affect alternative splicing. For example, some risk alleles in genes including CLU, PICALM and PTK2B exert their biological impacts through alternative splicing in the pathogenesis of AD (Raj et al., 2018). The 18 splice site variants that were predicted to cause alternative splicing in this study should be further investigated for their structural and functional impacts on the proteins.

Prediction analysis of sSNPs

Out of the 399 sSNPs we studied, eight sSNPs were predicted as deleterious sSNPs in PredictSNP2, including five from PICALM gene and three from SYNJ1 gene. Table 3 shows the eight deleterious sSNPs. Tables S5–S7 describe the complete prediction results of sSNPs in PICALM, SYNJ1 and SH3KBP1 genes, respectively.

Table 3 Deleterious sSNPs of PICALM and SYNJ1 genes predicted by PredictSNP2.

Variants ID	Ref/Alt	Minor allele frequency	CADD	DANN	FATHMM	FunSeq2	GWAVA	PredictSNP2	
PICALM									
rs779752380	C/G	0.000010/1	15.54	0.947688	0.91404	4	0.37	0.549842	
rs777199260	A/G	0.000008/1	8.879	0.869109	0.93421	4	0.42	0.091749	
rs775830584	T/C	NA	13.32	0.963935	0.21828	3	0.32	0.111345	
rs759002840	G/A	0.0002/20	17.47	0.964966	0.79634	0	0.34	0.459234	
rs367839126	G/A	0.00003/3	19.81	0.959689	0.889	0	0.41	0.485786	
SYNJ1									
rs768565296	C/T	0.000010/1	17.88	0.91747	0.91944	0	0.58	0.473949	
rs869206401	G/A	NA	22.4	0.934833	0.75779	0	0.51	0.028194	
rs772641102	G/A	NA	21.4	0.942502	0.36483	0	0.41	0.019858	

Among the 99 sSNPs in PICALM gene, five were predicted to have deleterious effects. Four out of those five, including rs779752380, rs777199260, rs775830584 and rs759002840, are located on the PICALM ANTH domain. Among the three deleterious sSNPs in SYNJ1 gene, only rs768565296 is located on the 5-phosphatase domain of the protein. For SH3KBP1 gene, all the 112 sSNPs were predicted to have neutral effects on the gene.

sSNPs affect protein expression and conformation through the regulation of RNA splicing, mRNA stability, translation dynamics, translation rate and co-translational folding of the protein (Sauna & Kimchi-Sarfaty, 2011). The actual biological impacts of the eight deleterious sSNPs predicted in this study should be validated in the future.

Prediction analysis of nsSNPs

All 759 nsSNPs in the three genes were analyzed by five prediction packages. The Ensembl genome database contains prediction results from SIFT and PolyPhen-2, in which a total of 106 nsSNPs were predicted as “damaging” in SIFT and “probably damaging” in PolyPhen-2. Together with the prediction results from Mutation Assessor, I-Mutant2.0 and SNPs&GO, a total of 18 nsSNPs exceeded the cutoff points of all five packages and were identified as deleterious nsSNPs. The prediction results of the deleterious nsSNPs are summarized in Table 4 while the complete prediction results of nsSNPs using five prediction packages are shown in Table S8. All the deleterious nsSNPs were predicted with high SIFT score and most of them have a PSIC score larger than 0.95 in PolyPhen-2. Prediction results from SIFT and PolyPhen-2 showed that all deleterious nsSNPs were highly conserved in the proteins (Table 4).

Table 4 Prediction of PICALM, SYNJ1 and SH3KBP1 deleterious nsSNPs.

Variant ID	AA Subs	Minor allele frequency	SIFT score	PolyPhen PSIC score	Mutation assessor	I-Mutant DDG	SNPs&GO	
PICALM								
rs750147583	L106S	NA	0	0.999	Medium	−1.93	0.644	
rs780443419	F109S	0.00001	0	0.994	Medium	−2.81	0.794	
rs145115354	D144N	0.000008	0	1	Medium	−1.57	0.784	
rs765338634	L179P	0.00001	0	0.989	Medium	−2.29	0.692	
SYNJ1								
rs781675993	N200K	0.000008	0	0.99	High	−1.34	0.735	
rs398122403	R258Q	0.00002	0	0.978	High	−1.23	0.841	
rs762909719	R289Q	0.000009	0	1	High	−1.77	0.723	
rs771755243	V338A	0.000008	0	0.928	Medium	−1.16	0.598	
rs768897710	Q414R	0.000008	0	0.999	High	−1.14	0.751	
rs779479360	G437D	0.00002	0	0.939	Medium	−1.42	0.736	
rs775515863	G487R	0.000008	0	0.988	Medium	−1.81	0.79	
rs752563697	G494D	0.000008	0	0.943	High	−1.74	0.632	
rs771070426	I515T	0.000008	0	0.969	Medium	−3.5	0.653	
rs756845805	G627S	0.000008	0	0.999	Medium	−1.6	0.751	
rs751110096	C723G	0.000008	0	0.996	Medium	−2.59	0.826	
rs147929290	I746T	0.000008	0.03	0.951	Medium	−2.42	0.633	
rs745418083	L776P	0.000008	0	0.981	Medium	−2.32	0.861	
SH3KBP1								
rs770229859	R648Q	NA	0	0.951	Medium	−1.07	0.658	

There were four deleterious nsSNPs in PICALM gene predicted in this study. All of them are located in amino acids 106 to 179 of the PICALM sequence. Positions 106 to 179 are located on the PICALM protein ANTH domain (23 to 283 aa) and ENTH domain (14 to 145 aa), and these two domains were reported to be functionally redundant (Maldonado-Baez et al., 2008; Manna et al., 2015). The ANTH domain of the protein is important in the binding to phosphatidylinositol 4,5-bisphosphate (PIP2). The binding of PICALM to PIP2 is important for the formation of clathrin-coated pit, which is one of the key functions of PICALM in CME (Ishikawa et al., 2015). These deleterious nsSNPs were predicted to cause conformation change and affect PICALM protein function. Fig. 1 shows the sticks representation of the protein structural changes caused by the deleterious nsSNPs in PICALM gene. In Fig. 1, the variant residues are colored yellow while red dashed lines indicate the hydrogen bonds between the residues. Variants rs780443419 (F109S) and rs765338634 (L179P) resulted in an addition or a deletion of hydrogen bond formation between the mutant and neighboring amino acids. Therefore, the substitution of these protein residues could significantly affect the ANTH domain function as well as the overall PICALM protein structure.

Figure 1 Structural comparison between the wild type and mutant amino acids of PICALM protein in sticks representation.

No addition or deletion of hydrogen bonds between the mutant residues and the surrounding residues were identified in variants (B) rs750147583 (L106S) and (F) rs145115354 (D144N), as compared to their wild type structures as shown in (A) and (E), respectively. (D) Variant rs780443419 (F109S) resulted in addition of hydrogen bond with residues L106, as compared to its wild type structure as shown in (C). (H) Variant rs765338634 (L179P) resulted in deletion of hydrogen bonds between residues M175 and D176, as compared to its wild type structure as shown in (G).

For SYNJ1 gene, 13 nsSNPs were predicted as deleterious. Six of them including rs781675993, rs398122403, rs762909719, rs771755243, rs768897710 and rs779479360 are located on the Sac domain while another four including rs756845805, rs751110096, rs147929290 and rs745418083 are located on the 5-phosphatase domain. Both Sac domain and 5-phosphatase domain mediate the crucial SYNJ1 protein function of phosphate hydrolysis, and they were reported to function in a concerted way (see review in Hsu & Mao (2013)). Besides that, a previous study has shown that both Sac and 5-phosphatase domains are important in the synaptic vesicle endocytosis and SYNJ1 D730A mutant completely eliminated the activity of 5-phosphastase domain (Mani et al., 2007). Although the actual impact of the predicted deleterious variants is yet to be determined, these variants may affect the domains function and influence the role of SYNJ1 in endocytosis as well.

Figure 2 Evolutionary conservation of the deleterious nsSNPs of PICALM, SYNJ1, and SH3KBP1 proteins by ConSurf server.

(A), (B) and (C) showed the ConSurf output for PICALM, SYNJ1, and SH3KBP1 protein sequences, respectively. The position of each deleterious variant is labeled with a black rectangle.

Variant rs770229859 is the only predicted deleterious nsSNP in SH3KBP1 gene. This variant changed residue arginine to glutamine at position 648, which is the C-terminal coiled-coil domain of the protein. The coiled-coil domain of SH3KBP1 was previously reported to be important in mediating the interaction between SH3KBP1 and phosphatidic acid (Zhang et al., 2009). Moreover, deletion of the coiled-coil domain directly affected the binding of SH3KBP1 to c-Cbl ubiquitin ligases, in which the interaction is essential for the initiation of CME regulated by SH3KBP1 (Zhang et al., 2009). Interestingly, another study has found that alanine mutation of four basic amino acids (K645, K646, R648 and R650) on the coiled-coil domain can completely disrupt the interaction of SH3KBP1 with phosphatidic acid and c-Cbl (Zheng, Zhang & Liao, 2014). Since the deleterious nsSNP rs770229859 resulted in amino acid substitution at position 648 of SH3KBP1, this variant is predicted to have a high likelihood of influencing the functions of the coiled-coil domain.

The structural and functional importance of the 18 deleterious nsSNPs were further analyzed using ConSurf analysis tools. Evolutionary conservation analysis determines the level of conservation of each protein residue and predicts the potential structural and functional importance of these deleterious variants to the protein. Figure 2 shows that 17 out of 18 (94%) deleterious nsSNPs were analyzed to be “conserved”, with 12 of them (70%) “highly conserved” (score “9”) through homologous sequence alignment. Only one deleterious nsSNP, rs745418083 (L776S) in SYNJ1 gene, was estimated to be “intermediate” in terms of evolutionary conservation. Besides that, 11 of the 18 (61%) deleterious variants were predicted as structural residues and the rest (39%) were functional residues. Figures S2–S4 shows the conservation scores of full length proteins of PICALM, SYNJ1 and SH3KBP1, respectively.

Besides the prediction analysis of the functional and structural importance of the deleterious nsSNPs and their level of conservation on the proteins, the changes of physical and chemical properties between wild type and mutant amino acids were studied. Table S9 shows the hydropathy, polarity and charge differences between the wild type and mutant amino acids of the deleterious nsSNPs.

During protein synthesis, both covalent and non-covalent interactions are driven by the physical and chemical properties of the amino acids. The hydropathy, size and charge of the amino acids all play important roles in the protein synthesis (Biro, 2006). Table S9 shows that the hydropathy in eight deleterious nsSNPs has changed from hydrophobic to hydrophilic, and the polarity of four nsSNPs has changed from non-polar to polar. The substitution of amino acids may affect both covalent and non-covalent interactions among amino acids, subsequently influencing the stability and conformation of protein structure.

To study the role of nsSNPs in affecting the total free energy and the stability of protein structures, full lengths of both wild type and mutant protein structures were modelled with the use of SPARKS-X server (Yang et al., 2011), and the structures were refined by using ModRefiner (Xu & Zhang, 2011). Energy minimization of the protein structures was performed by GROMOS96 force field using Swiss-PdbViewer. Table 5 shows the energy deviation and RMSD between wild type protein structures of PICALM, SYNJ1 and SH3KBP1, and the 18 mutant protein structures of deleterious nsSNPs. RMSD values were calculated to provide an insight into the deviation between the wild type and mutant protein structures. The RMSD values are proportional to the deviation between the wild type and mutant structures. The total free energy of wild type PICALM structure was −24,238.746 kJ/mol, and the total free energy of the four PICALM mutant structures was marginally decreased. The RMSD values also showed a minor deviation ranging from 1.13 to 1.45 Å. For SYNJ1 protein, the total free energy of the wild type structure was -19307.051 kJ/mol. The deviation of total free energy between SYNJ1 wild type and mutant protein structures is significant. Among the 13 SYNJ1 mutant structures, variant rs398122403 (R258Q) has the highest total free energy of 3,057.503 kJ/mol, and variant rs775515863 (G487R) has the lowest total free energy of −28,591.270 kJ/mol. Among all the SYNJ1 variants, rs781675993 (N200K) has the highest RMSD value of 2.34 Å. Figure 3 shows the structural comparison of the wild type SYNJ1 structure (green) and rs781675993 mutant structure (blue). Superposition of the two structures shows the structural deviation caused by the variant. The only SH3KBP1 mutant structure caused by variant rs770229859 (R648Q) showed a decreased total free energy of −1,6167.838 kJ/mol, with a RMSD value of 1.88 Å.

Table 5 Total energy deviation and RMSD of deleterious nsSNPs of PICALM, SYNJ1 and SH3KBP1 proteins.

Variant ID	AA Subs.	Total energy (kJ/mol)	Energy difference	RMSD (Å)	
PICALM (wild type)		−24,238.746			
rs750147583	L106S	−24,759.834	−521.088	1.26	
rs780443419	F109S	−24,921.291	−682.545	1.45	
rs145115354	D144N	−24,494.619	−255.873	1.45	
rs765338634	L179P	−24,824.025	−585.279	1.13	
SYNJ1 (wild type)		−19,307.051			
rs781675993	N200K	−11,031.447	+8,275.604	2.34	
rs398122403	R258Q	3,057.503	+22,364.554	2.15	
rs762909719	R289Q	−21,897.707	−2,590.656	2.02	
rs771755243	V338A	−17,907.066	+1,399.985	2.09	
rs768897710	Q414R	−18,169.162	+1,137.889	2.04	
rs779479360	G437D	−19,518.217	−211.166	2.08	
rs775515863	G487R	−28,591.270	−9,284.219	2.1	
rs752563697	G494D	−6,042.169	+13,264.882	2.24	
rs771070426	I515T	−9,617.264	+9,689.787	2.13	
rs756845805	G627S	−25,429.416	−6,122.365	2.13	
rs751110096	C723G	−23,460.332	−4,153.281	1.83	
rs147929290	I746T	−20,893.076	−1,586.025	2.25	
rs745418083	L776P	−18,299.359	+1,007.692	2.14	
SH3KBP1 (wild type)		−15,365.350			
rs770229859	R648Q	−16,167.838	−802.488	1.88	

Figure 3 Superposition of wild type SYNJ1 structure and variant rs781675993 (N200K) mutant structure.

This figure shows the truncated protein structure of 100 residues around the mutant sites. Wild type SYNJ1 structure is green, and mutant SYNJ1 structure is blue. Wild type residue (N) at position 200 is red, and mutant residue (K) at position 200 is purple.

Verification of nsSNP prediction principle

To demonstrate the reliability of nsSNP prediction, we predicted the functional consequences of other nsSNPs that have been previously studied in other benchwork experiments. Ten PSEN1 pathogenic nsSNPs that have been validated experimentally to affect amyloid-beta (Aβ) level were retrieved from ALZFORUM (https://www.alzforum.org/). Besides that, another five PSEN1 nsSNPs, including three non-pathogenic variants and two variants that have never been reported to be deleterious or disease-associated, were selected as negative controls. The prediction results of the total 15 nsSNPs in PSEN1 gene are shown in Table S10. The prediction results show that nine out of these ten pathogenic nsSNPs were predicted deleterious by all five prediction packages used in this paper. The only pathogenic nsSNP that was not predicted as deleterious variant, which is rs63750231 (E280A), has I-Mutant DDG and SNPs&GO score that is lower than the cutoff point. All these five negative controls of PSEN1 nsSNPs were predicted to be non-deleterious to the protein.

Computational prediction of the biological consequences of variants was previously reported to yield a high false positive (FP) and low false negative (FN) rate, suggesting that computational prediction tends to overestimate the deleterious effects of variants (Miosge et al., 2015). To reduce the FP rate, our study implemented meta-prediction method to perform functional prediction of nsSNPs. Despite one FN prediction result (rs63750231) was reported, the overall prediction results of the PSEN1 pathogenic and non-pathogenic nsSNPs indicate that the prediction approaches which were used, reliably predicted the functional consequences of nsSNPs.

NGS data of AD studies

To check if any of the 56 deleterious variants were identified in the next generation sequencing (NGS) studies of AD, genotype calling data (variant calling format (.VCF) files) were retrieved from three studies including Alzheimer’s Disease Sequencing Project (ADSP), Alzheimer’s Disease Neuroimaging Initiative (ADNI) and Mount Sinai Brain Bank (MSBB) studies. ADSP aims to identify genetic variants that increase the risk for or protect against AD. The study consists of two parts, including a whole genome sequencing (WGS) of 584 samples family-based study and a whole exome sequencing (WES) of 10,061 samples case-control study (Beecham et al., 2017). Genotype calling data of both WGS and WES were retrieved from the study. For ADNI, the study is primarily designed to discover novel biomarkers from a total of 800 individuals for the early detection of AD. The 800 individuals consist of 200 elderly controls, 400 mild cognitive impairment (MCI) and 200 AD patients. WGS of the 800 individuals was performed in the following add-on study and the genotype calling data is publicly available (Weiner et al., 2015). For MSBB study, it aims to discover novel molecular mechanism of AD pathogenesis from genomic, transcriptomic and proteomic data consisting of 364 AD case-control human brains (Wang et al., 2018). The raw and processed data of MSBB study are available through the Synapse software platform (https://www.synapse.org/#!Synapse:syn3159438) and the .VCF file of WES of the study was retrieved.

Among the 56 deleterious variants predicted in this study, seven variants were also identified in ADSP, ADNI or MSBB studies. Table S11 shows the deleterious variants that were identified in the NGS studies of AD. Identification of these deleterious variants in these NGS studies further suggests their potential association with AD. In these studies, the genotype data was mapped on the reference genome GRCh37 while the variant data retrieved from Ensembl genome database was mapped on GRCh38. Therefore, the variant positions in both GRCh37 and GRCh38 are provided in Table S11.

Discussion

In this study, a total of 56 SNPs in PICALM, SYNJ1 and SH3KBP1 genes were predicted as deleterious variants for the genes and their gene products. These deleterious variants included 12 UTR variants, 18 splice site variants, 8 sSNPs and 18 nsSNPs. These 12 deleterious UTR variants were predicted to cause an addition or a deletion of cis-acting elements in UTR sequences, which may affect the post-transcriptional regulation of the genes. These 18 deleterious splice site variants were predicted to disrupt gene splicing, which can produce truncated or extended forms of the proteins. Meta-analysis by PredictSNP2 has predicted a total of 8 deleterious sSNPs in PICALM and SYNJ1 genes. These deleterious UTR variants, splice site variants and sSNPs were expected to affect the regulation of gene expression, as well as the stability of the mRNAs. For the 18 deleterious nsSNPs, most of them are located on the functional domains of the proteins. These domains mediate the main protein functions in the regulation of clathrin-mediated endocytosis (CME). ANTH domain on PICALM regulates the formation of clathrin-coated pit while coiled-coil domain on SH3KBP1 has an important role in the initiation of CME (Ishikawa et al., 2015; Zhang et al., 2009). Specifically, both of the domains are involved in the early stage of CME. For SYNJ1, the hydrolysis of PIP2 by Sac domain and 5-phosphatase domain may be involved in the disassembly of clathrin-coated vesicles (Hsu & Mao, 2013), which is in the late stage of CME. Therefore, identification of deleterious nsSNPs on these functional domains may not only help to elucidate the functional impacts of these variants to the proteins, but also may gain insight into the biological roles of these domains in CME. Moreover, three endocytosis genes in this study have been linked to the pathogenesis of AD. Hence, the prediction analysis of the deleterious variants may be relevant for the gene-disease association study.

To date, GWAS and other genetic studies have successfully identified more than 20 risk genes associated with AD, but the molecular mechanisms of these genes remain to be determined. To understand the molecular basis of these AD-associated genes in the disease, one of the approaches is to identify functional SNPs from a vast number of neutral SNPs in the gene of interest. However, it was estimated that SNP occurs in every 100 to 300 bases and there are up to ten million SNPs in the entire human genome (Ke, Taylor & Cardon, 2008). The massive number of SNPs in the entire human genome makes it impractical to perform laboratory experiments to determine the functional consequences of all the SNPs, even for a single gene. For that reason, computational methods become an alternative and important way to prioritize the SNPs that are possibly structurally or functionally significant for the genes of interest. Computational methods such as prediction and modelling tools allow researchers to identify functionally significant SNPs from neutral SNPs. The prediction accuracy is expected to be improved when results from multiple algorithms are combined to perform meta-prediction. Besides that, computational methods are able to provide high throughput prediction results at lower cost and faster time, as compared with laboratory experiments. Although study by Miosge et al. (2015) has pointed out the low specificity of computational prediction, the discrepancy between the prediction and experimental results of the variants impacts can be further improved by providing a better data quality and input alignment design during computational prediction (Gallion et al., 2017). In conclusion, computational methods is an essential part to provide initial assessment and prioritize the SNPs that are most likely to be functional, which can largely improve research efficiency.

Although computational methods can significantly contribute to the task of identifying functional SNPs, a limited number of studies had performed function prediction analysis of the variants located in the disease-associated genes. Moreover, most of the function prediction studies of variants only focus on annotating the functional consequences and disease-relatedness of nsSNP since it was conventionally thought to have a higher impact on the phenotype rather than the other types of variations. However, nsSNPs, sSNPs and 5′UTR variants have been reported to share similar likelihoods of association in Mendelian diseases (Chen et al., 2010). Other than these three types of variants, more than 18,000 of all variants reported in the Human Gene Mutation Database (HGMD) are splicing variants, which is about 9% of the total number of variants in the database (Stenson et al., 2017). In the pathogenesis of complex diseases, the roles of genetic factors are much more complicated than those in Mendelian diseases. Genetic variants in non-coding regions may play a more prominent role than in coding regions in complex diseases since most of the disease-associated variants identified in GWAS are located in non-coding regions (Zhang & Lupski, 2015). Identifying pathogenic variants in both coding and non-coding regions of genes associated with complex diseases can help to explain the genetic factors and networks of the diseases. Therefore, this study does not target nsSNPs exclusively because it may overlook other functionally significant variants that are associated with AD.

To our knowledge, 55 out of the 56 deleterious variants predicted in this study were reported for the first time to be functionally or structurally significant for PICALM, SYNJ1 and SH3KBP1 genes, and their gene products. Only one deleterious variant, which is rs398122403 (R258Q) in SYNJ1 gene, was previously reported to be associated with Parkinson’s disease (PD) (Krebs et al., 2013). Variant rs398122403 was found to disrupt the phosphatase activity of Sac domain on SYNJ1 protein, leading to an impaired synaptic function of SYNJ1 and causing an accumulation of endocytic proteins in the brains of mutant mice, developing neurological symptoms similar to PD patients (Cao et al., 2017). Although AD and PD have different clinical and pathological symptoms, many studies suggested that they share some common molecular mechanisms, such as oxidative stress and mitochondrial dysfunction (see review in Tan et al. (2019)). However, in the case of SYNJ1, several studies suggested that the downregulation of SYNJ1 may alleviate the pathogenesis of AD (McIntire et al., 2012; Zhu et al., 2013). Intriguingly, the biological effects of the SYNJ1 in AD don’t seem to be in line with budding yeast and C. elegans orthologues (Treusch et al., 2011). Since the validated PD-associated variant rs398122403 was also predicted as deleterious variant in this study, and the mutant protein structure of this variant has the highest total free energy among all the deleterious variants, the prediction results of this study may serve as a good target for further investigation. Nevertheless, it is not known if the variant rs398122403 is also associated with AD. SYNJ1 is one of very few proteins associated with both AD and PD. Therefore, the SYNJ1 variants predicted in this study may be useful to discover the biological roles of SYNJ1 on the molecular mechanisms in AD and PD.

Previously, several AD-associated common variants have been identified from GWAS. For example, variants rs3851179, rs541458 and rs561655 are associated with AD. These three variants were not identified in this study since they are intergenic variants which are located upstream of PICALM gene. On the other hand, the minor allele frequency (MAF) of these three SNPs are over 0.3, which is much higher as compared to the deleterious variants predicted in this study. Previous studies have reported that most of the AD-associated common variants have a smaller effect size on the disease risk as compared to rare variants (see review in Nicolas, Charbonnier & Campion (2016)). Therefore, the current research direction is focused on identification of rare variants with a larger effect size that are located in different genes (Pierre & Genin, 2014). The MAF of the 56 deleterious variants predicted in this study are provided in Tables 1–4. The MAF data was obtained from the Exome Aggregation Consortium (ExAC) in aggregated populations. All the deleterious variants are very rare variants (average MAF < 0.00002). According to the “common disease common variant” (CDCV) hypothesis, the risk of getting a common disease would be relatively high along with the appearance of disease-predisposing alleles. Assuming that the common alleles are the actual cause of common diseases, many causative alleles would have been identified through GWAS. However, the current data do not seem to support this hypothesis since only a very limited number of variants have been successfully linked to the pathogenesis of certain diseases. Moreover, common variants identified from GWAS are insufficient to explain the missing heritability of complex phenotypes and diseases (see review in Blanco-Gómez et al. (2016)). On the other hand, the rare variants absent in GWAS have been suggested to be an important factor in common disease susceptibility in the common disease rare variant (CDRV) hypothesis (Auer & Lettre, 2015). Despite different strategies used in the identification of disease-associated variants in both hypotheses, they can work together to identify the common variants along with the genomic region and locus through GWAS, followed by conducting the functional analysis of the rare variants. Currently, no AD-associated rare variant in PICALM, SYNJ1 and SH3KBP1 genes has been reported by other studies so far. It is plausible that the sample size of GWAS might not be sufficient to identify disease-associated rare variants. Hence, our prediction analysis may be useful to identify functional variants which could be absent in GWAS due to their rare MAF.

Conclusion

In our study, a total of 56 rare variants in PICALM, SYNJ1 and SH3KBP1 genes were predicted as deleterious variants. These deleterious variants were predicted to affect the regulation of gene expressions and protein functions. These genes have cellular functions involved in clathrin-mediated endocytosis (CME) and deleterious variants in these genes were expected to affect the functions of these proteins in endocytosis. Moreover, these genes were previously reported as AD-associated and they are implicated in the pathogenesis of AD. The deleterious variants in these genes may affect AD pathogenesis through the disruption of the gene expressions and protein functions. In addition, seven of the deleterious variants were also identified in three large NGS studies of AD, further suggesting their possible roles in AD pathogenesis. The actual roles of these deleterious variants in affecting protein functions in CME, as well as their implications in AD, are recommended to be validated by future laboratory experiments.

Supplemental Information

Figure S1 Total number of variants of PICALM, SYNJ1 and SH3KBP1 genes

Click here for additional data file.

Figure S2 Evolutionary conservation of PICALM protein sequence

The result file was generated by ConSurf server.

Click here for additional data file.

Figure S3 Evolutionary conservation of SYNJ1 protein sequence

Click here for additional data file.

Figure S4 Evolutionary conservation of SH3KBP1 protein sequence

The result file was generated by ConSurf server.

Click here for additional data file.

Table S1 Possible consequences of splice site variants predicted by MaxEntScan

Click here for additional data file.

Table S2 UTR sequences information and the total number of variants of PICALM, SYNJ1 and SH3KBP1 genes

Click here for additional data file.

Table S3 Prediction analysis of UTRs variants

S3A-S3C provided the complete prediction result of PICALM, SYNJ1 and SH3KBP1 UTR variants, respectively. S3D showed the list of variants that changed the number of motifs as compare to the respective wild type sequences.

Click here for additional data file.

Table S4 Prediction analysis of splice site variants

S4A-S4C provided the complete prediction result of PICALM, SYNJ1 and SH3KBP1 splice site variants, respectively. S4D showed the list of variants with score difference exceeding the defined threshold, which were predicted to break or create splice site on the respective genes.

Click here for additional data file.

Table S5 Prediction analysis of PICALM sSNPs

The result file was generated by PredictSNP2.

Click here for additional data file.

Table S6 Prediction analysis of SYNJ1 sSNPs

The result file was generated by PredictSNP2.

Click here for additional data file.

Table S7 Prediction analysis of SH3KBP1 sSNPs

The result file was generated by PredictSNP2.

Click here for additional data file.

Table S8 Prediction analysis of nsSNPs

S8A-S8C provided the complete prediction result of PICALM, SYNJ1 and SH3KBP1 nsSNPs, respectively. S8D showed the list of variants that were predicted to be the deleterious variants in this study.

Click here for additional data file.

Table S9 The physical and chemical properties of deleterious nsSNPs of PICALM, SYNJ1 and SH3KBP1 proteins

Click here for additional data file.

Table S10 Prediction of pathogenic and non-pathogenic PSEN1 nsSNPs

Click here for additional data file.

Table S11 Deleterious variants that were identified in the NGS studies of AD

Click here for additional data file.

Data S1 Raw data of UTR variants

The information of PICALM, SYNJ1 and SH3KBP1 UTR variants were retrieved from Ensembl genome database.

Click here for additional data file.

Data S2 Raw data of splice site variants

The information of PICALM, SYNJ1 and SH3KBP1 splice site variants were retrieved from Ensembl genome database.

Click here for additional data file.

Data S3 Raw data of sSNP variants

The information of PICALM, SYNJ1 and SH3KBP1 sSNP variants were retrieved from Ensembl genome database.

Click here for additional data file.

Data S4 Raw data of nsSNP variants

The information of PICALM, SYNJ1 and SH3KBP1 nsSNP variants were retrieved from Ensembl genome database.

Click here for additional data file.

We would like to show our gratitude to Dr. Henry Lee Seldon and Dr. Vachiranee Limviphuvadh for their valuable opinions to improve the manuscript.

Additional Information and Declarations

Competing Interests

Author Contributions

Data Availability

The authors declare there are no competing interests.

Han Jieh Tey conceived and designed the experiments, performed the experiments, analyzed the data, prepared figures and/or tables, authored or reviewed drafts of the paper.

Chong Han Ng conceived and designed the experiments, contributed reagents/materials/analysis tools, authored or reviewed drafts of the paper, approved the final draft.

The following information was supplied regarding data availability:

The raw data for prediction and analysis of UTR variants are available in Data S1. The complete prediction results of these raw data are available in Table S3 and the results are summarized in Table 1.

The raw data for prediction and analysis of splice site variants are available in Data S2. The complete prediction results of these raw data are available in Table S4 and the results are summarized in Table 2.

The raw data for prediction and analysis of sSNPs are available in Data S3. The complete prediction results of these raw data are available in Tables S5–S7 and the results are summarized in Table 3.

The raw data for prediction and analysis of nsSNPs are available in Data S4. The complete prediction results of these raw data are available in Tables S8 and summarized in Table 4.

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
