# Peer review of "Computational analysis of functional SNPs in Alzheimer’s disease-associated endocytosis genes"

_PeerJ, doi:10.7717/peerj.7667_

## Round 0.1 · original submission · Major Revisions

Both reviews are generally positive about the paper. However, multiple comments are made. Please follow the reveiwers's suggestions to revise.

·

Basic reporting

Level of English is high. I've found just a few typos:
Line 29: were further analyzed → was further analyzed
Line 39: may implicated → may be implicated
Line 46: with prevalence → with a prevalence of
Line 109: to have higher impact on → to have a higher impact on
Table 4: Headline is doubled (first two rows are the same).

Literature references are sufficient with an exception of linking R258Q variant to PD.(further explained in section 3).

It’s well-known now that there is no “junk DNA” and variants outside protein-coding regions might play a crucial role in many diseases. However, referring to the certain numbers (line 111; 1.26% and 1.46% for sSNPs & nsSNPs) seems somehow redundant & misleading without further explanation. It differs a lot between types of diseases (mendelian versus complex) and such quantification is very tricky. I would recommend rewriting this part with using more recent sources (again, numbers are not necessary).

In general, I would appreciate to better connect the general explanation of types of variants and their causal disease-causing effect with AD, not with chronic hepatitis B infection or familial dysautonomia (paragraph between lines 118-125).

Experimental design

I would like to see a better explanation of why the authors decided to limit their analysis to SYNJ1, PICALM and SH3KBP1. Late-Onset AD risk endocytosis-related genes involve also some other ones, e.g, BIN1, CD2AP, EPHA1, and SORL1.

The Methods section is clearly written. I’ve carefully reviewed the selection & settings of all predictors: authors seem to follow recommended settings & best practices or set their own ones in a very reasonable way. All the necessary details are provided.

Validity of the findings

The authors did not report that one of 56 highlighted variants (R258Q in SYNJ1) is strongly associated with Early-onset Parkinson Disease [PMID: 28331029]. The disease-causing effect is explained by the loss of dephosphorylation activity of Sac1 domain. It’s encouraging that this literature-validated variant introduced the highest total free energy in authors' simulations and keeps a promise that the other variants might be good targets for further investigation as well. The authors should check literature and discuss this variant properly in the manuscript.

Regarding the validation of highlighted variants in large cohorts from sequencing: I suggest to include also AD-control MSBB cohort (part of AMP-AD) – getting access is relatively easy and can help to pre-validate more variants (optional):
https://www.nature.com/articles/sdata2018185 (follow links to Synapse data as Data Citations)

Additional comments

The manuscript is clearly written and the topic is of high importance. In my opinion, the results are worth publishing. There are several areas, where I would suggest some improvements.

·

Basic reporting

Overall satisfactory but with deficits in context integration with this study.

- Clear and unambiguous, professional English used throughout. Some grammatical and syntactical errors are found dispersed in the text, so authors may want to go over the document one more time, examples provided below:
L266: disorder -> disorders
L492: involve -> be involved
Table4_L6: (continue) -> (continued)

- Literature references, sufficient field background/context provided. This is where I feel the manuscript requires significant improvement as follows:
1) Regarding the proteins chosen for the study, in the introduction, it should be more clearly stated that the 3 proteins that were selected were the ones that either classified as “Validated risk factor” or “Interacts with validated risk factor…” in PMID: 22033521
2) Regarding SYNJ1, since this study tries to relate the results with AD, I feel that the authors need to improve the introduction, results and discussion section to address the following issues: Although in PMID: 22033521 SYNJ1 is a yeast Aβ suppressor, in a series of studies from the Cai group (PMID: 24052255, 26372964, 28900205) some of which are cited by the authors, it seems that in rodents SYNJ1 may be an Aβ enhancer or may increase pTau especially secondary to certain stress conditions such as TBI. In addition, mutations that selectively abolish Sac1 function lead to early onset parkinsonism whereas critical reduction of the dual phosphatase activity may be linked with neurodegeneration (maybe it would be helpful for authors to read PMID:27435091 for context). All these studies have to be discussed because they affect the definition of deleterious mutations in the context of AD. Authors to ensure that similar issues are not taking place with PICALM and SH3KBP1
3) L332-4: Please provide reference supporting the statement that sSNPs can result in protein misfolding
4) L363-4: Please explain why this is important “Furthermore, a portion… …in AD patients.”

- Professional article structure, figures, tables. Raw data shared: Good except minor inconsistency: either put “table… (continued)” in all tables that span multiple pages or in none.

- Self-contained with relevant results to hypotheses: Yes

Experimental design

Study is very good in this regard; minor improvement suggestions are given below.

- Original primary research within Aims and Scope of the journal: Yes

- Research question well defined, relevant & meaningful. It is stated how research fills an identified knowledge gap: Yes

- Rigorous investigation performed to a high technical & ethical standard: Yes

- Methods described with sufficient detail & information to replicate: Clearly written with all information required, except as below:
1) L438-449: The authors have included a nice validation for their approach. On the other hand, authors should clearly describe how they selected the 10 PSEN1 nsSNPs for AD. There are literally hundreds more and a lot of them have been more conclusively linked with AD (e.g. E280A to name one). Similarly, how they selected the random 5 not pathogenic. If applicable, the authors should consider including the rs63750592, rs63750771, and rs112451138 as they have the highest level of confidence of not being pathogenic.
2) L280-3 Authors mention critical role of miRNAs for regulating gene expression via 3’UTR but they haven’t performed a miRNA binding site prediction specific analysis (with e.g. TargetScan) for these genes. Maybe authors could comment that such an analysis was outside the scope of this study in the discussion.

Validity of the findings

There is good support for findings

- Data is robust, statistically sound, & controlled: Yes

- Conclusions are well stated, linked to original research question & limited to supporting results: Yes except as for SYNJ1 as related to the context described above.

Additional comments

The manuscript is clearly written, and it can be a good resource for the potential role of SNPs for these genes that have been implicated in AD. As, I have mentioned above the biggest deficits are in presenting findings in the context of existing literature and in relevance with AD.

---

## Round 0.2 · accepted · Accept

Please double check a few language issues pointed out by reviewer 2:

There are some minor language issues that could use some work (e.g. l90: This -> These). In addition, there is one sentence that doesn't read well although the meaning is somewhat conveyed (first sentence in l257-8): "The biological consequences of alternative splicing are frequently affected by genetic variations".

·

Basic reporting

Language has been significantly improved compared to the previous version but minor mistakes were introduced in the newly added parts. These could be addressed by the editorial team (no need for an additional round of revision).

Experimental design

No comment.

Validity of the findings

No comment.

Additional comments

Thank you for addressing my concerns in the revised manuscript.